# A Self-Powered, Skin Adhesive, and Flexible Human–Machine Interface Based on Triboelectric Nanogenerator

**DOI:** 10.3390/nano14161365

**Published:** 2024-08-20

**Authors:** Xujie Wu, Ziyi Yang, Yu Dong, Lijing Teng, Dan Li, Hang Han, Simian Zhu, Xiaomin Sun, Zhu Zeng, Xiangyu Zeng, Qiang Zheng

**Affiliations:** 1Engineering Research Center of Intelligent Materials and Advanced Medical Devices, School of Biology and Engineering, Guizhou Medical University, Guian New District, Guiyang 561113, China; jalaaron77@gmail.com (X.W.); zi1yang412@gmail.com (Z.Y.); dongyu898@gmail.com (Y.D.); ljteng@gmc.edu.cn (L.T.); lid825266@gmail.com (D.L.); hanhang209@gmail.com (H.H.); simianzhu@gmc.edu.cn (S.Z.); sunxiaomin@gmc.edu.cn (X.S.); 2Key Laboratory of Infectious Immune and Antibody Engineering of Guizhou Province, Engineering Research Center of Cellular Immunotherapy of Guizhou Province, School of Biology and Engineering/School of Basic Medical Sciences, Guizhou Medical University, Guian New District, Guiyang 561113, China; 3Immune Cells and Antibody Engineering Research Center of Guizhou Province, Key Laboratory of Biology and Medical Engineering, Guizhou Medical University, Guian New District, Guiyang 561113, China

**Keywords:** triboelectric nanogenerators, human–machine interaction, self-powered, flexible, skin adhesive

## Abstract

Human–machine interactions (HMIs) have penetrated into various academic and industrial fields, such as robotics, virtual reality, and wearable electronics. However, the practical application of most human–machine interfaces faces notable obstacles due to their complex structure and materials, high power consumption, limited effective skin adhesion, and high cost. Herein, we report a self-powered, skin adhesive, and flexible human–machine interface based on a triboelectric nanogenerator (SSFHMI). Characterized by its simple structure and low cost, the SSFHMI can easily convert touch stimuli into a stable electrical signal at the trigger pressure from a finger touch, without requiring an external power supply. A skeleton spacer has been specially designed in order to increase the stability and homogeneity of the output signals of each TENG unit and prevent crosstalk between them. Moreover, we constructed a hydrogel adhesive interface with skin-adhesive properties to adapt to easy wear on complex human body surfaces. By integrating the SSFHMI with a microcontroller, a programmable touch operation platform has been constructed that is capable of multiple interactions. These include medical calling, music media playback, security unlocking, and electronic piano playing. This self-powered, cost-effective SSFHMI holds potential relevance for the next generation of highly integrated and sustainable portable smart electronic products and applications.

## 1. Introduction

The advances in flexible wearable sensors have led to diverse applications [1,2,3,4]. Among these applications, human–machine interactions (HMIs) are increasingly paramount, which effectively connect human intentions with machine operations [5,6]. Nowadays, most of the reported flexible wearable sensors utilized in HMI are developed based on the mechanisms of resistive or capacitive [7,8]. However, resistive and capacitive sensors require a continuous power supply, which greatly increases the power consumption of the whole system. Since the first invention in 2012 by Prof. Z. L. Wang and his team [9], the triboelectric nanogenerator (TENG) offers unique advantages such as flexibility, scalability, and self-powered operation, making them ideal candidates for sensing in the HMI [10,11].

In the past few years, there has been significant progress in the research and development of TENG-based HMI systems. In order to meet diverse human–machine interaction needs, efforts have been intensified to develop multifunctional TENG sensors capable of detecting different types of physical stimuli [12,13,14,15,16,17,18,19,20]. This involves ensuring their comfort and fit, along with reliability and self-powered capabilities. To achieve these aims, the materials and structures of the TENG are specially designed [21,22,23,24,25,26]. Although these efforts have improved the performance of TENG-based HMI systems to a certain extent, they inevitably increase the complexity and difficulty associated with fabricating and miniaturizing the system.

On the other hand, the accuracy and stability of signal recognition are crucial in HMI applications. However, most of the TENG sensors can be easily affected by interference from the surrounding environment, which may result in instability and misidentification [27]. In this regard, efforts have been made to reduce the interference, including designing electronic circuits and algorithms to filter the noise, and synthesizing special materials to serve as the packaging structure of the TENG to isolate interfering electrical signals [28,29,30,31,32]. Obviously, these methods require extra advanced materials or equipment, which increases the cost of HMI applications.

In addition, the tissue adhesion interface is important for HMI systems to adhere to the complex human skin surface, which can help to maintain the stability of use. Hydrogel is conducive to adhesion to different substrates, which is suitable to be used as a medium for the HMI device to adhere to human skin, and the common materials and preparation methods of hydrogel are easy to realize [33,34,35,36,37].

Here in this work, we report a self-powered, skin adhesive, and flexible human–machine interface (SSFHMI) for wearable human–machine interaction (HMI). Compared with previously established TENG-based HMI interfaces [30,32,37], the SSFHMI sets itself apart with a more simplistic fabrication process and a lower rate of signal misinterpretation. A single TENG unit, with a contact area of 1 cm^2^, is capable of reaching an output voltage of 13 V and a short-circuit current of 12 nA, an output power density of 0.29 mW/m^2^ when connected to an external load with a resistance of 50 MΩ. After 18,000 cycles of contact separation, there is no significant decay in the output. Therefore, it can be easily integrated into a microcontroller to serve as a stable trigger signal. Moreover, a hydrogel is synthesized to serve as adhesive interface, the adhesive strength reaches up to 20 kPa between the SSFHMI and tissue, ensuring a strong adhesion between the device and the human body surface. In addition, we prepared a flexible skeleton spacer to prevent the electrical signals interfering with each other. Finally, by encoding the electrical signals, we demonstrate several typical application scenarios of the proposed SSFHMI, including medical calling, music media playback, security unlocking, and electronic piano playing. The SSFHMI could potentially be extended to other fields, including smart wear, information security, virtual reality (VR), and augmented reality (AR), among others.

## 2. Materials and Methods

### 2.1. Preparation of PDMS Film with Microstructure

A 3D printer (Bambu Lab P1S, Shenzhen, China) is used to print a groove mold with a depth of 200 μm, and the bottom of the groove is arranged with regular bumps, forming a wavy pattern at the bottom. The main agent and curing agent are mixed at a ratio of 10:1, stirred for 10 min, absorbed into the mold with a babbitt straw and wiped flat, then put it into a vacuum antifoam bucket and vacuumed for 10 min. Then put it into a blast-drying oven (DHG-924385-III, Shanghai Cimo Medical Instrument Co., Ltd., Shanghai, China), set the temperature in the oven to 50 °C, and crosslink for 30 min.

### 2.2. Fabrication of SSFHMI Based on TENG

Firstly, the microstructured PDMS film is cut and integrated onto a flexible FPCB board that incorporates circuits and electrodes. Subsequently, a skeleton spacer with nine square holes is fabricated using a 3D printer (Bambu Lab P1S, Shenzhen, China) and adhered to the surface of the flexible FPCB. Finally, polytetrafluoroethylene (PTFE) is adhered to the skeleton spacer. The FPCB used in this design was customized by the Ruida Express Circuit Board Company (Shenzhen, China).

### 2.3. Characterization of the Electrical Performance of the Device

The output performance of a single-electrode TENG is measured and recorded using a linear motor (Y400TA100-600, Beijing Hangtian Jingyi Technology Co., Ltd., Beijing, China), an electrometer (6517B, Keithley, Keithley Instruments, Cleveland, OH, USA), resistors of different resistance values, a multimeter (DMM6500, Keithley), and a four-channel oscilloscope (MSO44, Tektronix, Beaverton, OR, USA). The output performance of the TENG array is measured and recorded in multiple paths using a system switch (3706A-S, Keithley) and an eight-channel oscilloscope (MSO58B, Tektronix).

The pressure–output voltage relationship of a sensing unit on the SSFHMI is measured and recorded using a tensile tester (F305, MARK-10, Copiague, NY, USA) and a four-channel oscilloscope (MSO44, Tektronix, Beaverton, OR, USA). A sensing unit is connected to the MSO44, and the output waveform on the MSO44 is observed and recorded after continuous, varying amplitude pressure is applied to the single-sensing unit using F305.

### 2.4. Design of the Processing Circuit in Human–Machine Interaction Applications

In this design, an Arduino UNO R3 (Somerville, MA, USA) is used as the signal acquisition and processing device. The built-in ADC of the Arduino UNO R3 is used to set the threshold of the trigger source. The judgment is made by writing a program to compare the size of the input signal and the threshold. Then, the corresponding instructions are set and executed to the host computer through serial port transmission.

### 2.5. Synthesis of P(OEGMA-co-DEGMA)/LAP Nanocomposite Hydrogel Adhesives

#### 2.5.1. Materials

Oligo (ethylene glycol) methyl ether methacrylate (OEGMA, Mn 500), di (ethylene glycol) methyl ether methacrylate (DEGMA, Mn 188), potassium persulfate (KPS, 99%), and N,N,N’N’-tetramethylethylenediamine (TEMED, 99%) were used as purchased (Sigma-Aldrich, St. Louis, MO, USA) without further purification. Laponite^®^ nanoclay (XLG, LAP) was purchased from BYK (Wesel, Germany).

#### 2.5.2. Synthesis Process

P(OEGMA-co-DEGMA)/LAP nanocomposite hydrogel adhesives were prepared through one-pot free radical copolymerization using OEGMA and DEGMA as monomers, and LAP as a physical crosslinker [38]. Firstly, 120 mg of LAP were dissolved in 4.5 mL of deionized water to obtain homogeneous LAP dispersions via magnetic stirring for 3 h. Secondly, 919 mg of DEGMA and 581 mg of OEGMA were added into the reaction mixture in an ice bath. Subsequently, 12 mg of KPS were added as a reaction initiator, and the aqueous mixture was degassed with nitrogen for 5 min, then 12 μL of accelerator TEMED was added to the solution. Finally, the mixture solution was injected into a 25 mm × 25 mm Teflon mold for polymerization at 25 °C for 24 h.

#### 2.5.3. Adhesive Strength Test and Self-Healing Capability Test of Nanocomposite Hydrogel Adhesive

In accordance with the modified ASTM F225505 standard [39], the adhesive strength of nanocomposite hydrogel adhesives to porcine skin tissues and the FPCB substrate were evaluated using the lap shear test. Porcine skin tissues were cut into 20 mm wide and 60 mm length. Then, a square nanocomposite hydrogel with a side of length 20 mm was sandwiched between two porcine skin tissues. After that, the sample was then withdrawn at a rate of 25 mm/min and the stress-displacement curve was obtained using a tensile tester (F305, MARK-10). Place the nanocomposite hydrogel adhesive on the rheometer (MCR 302e, Anton Paar, Graz, Austria), record its storage modulus and loss modulus, and plot the graphs. Characterize its self-healing ability through the graphical data.

## 3. Results and Discussion

### 3.1. Structure and Working Principle of the SSFHMI

As schematically illustrated in Figure 1a, in the process of crafting this SSFHMI, we meticulously amalgamated nine units of 1 × 1 cm^2^ triboelectric nanogenerators (TENGs) into a 3 × 3 array, housed on a flexible printed circuit board (FPCB). The signals procured from the TENG matrix, following transmission through a distinct circuit, are conveyed to a microcontroller. It is upon this process that the identification of the signal origins occurs, leading to the initiation of appropriate responsive commands. These commands efficiently cater to a cornucopia of applications, such as medical calls, media player control, password unlocking, and electronic keyboard playing. After the acquisition of an external touch signal, the device exhibits the capability to transduce the physical interaction into electrical signals. Once these signals have undergone the necessary processing, they are channeled into the microcontroller; the resultant operations are then represented visually on a computing interface.

The construction of the interface can generally be divided into three parts: base layer, circuit, and TENG. As shown in Figure 1b, the base layer can be further divided into an adhesive layer and a basal layer. The layout circuit of the basal layer is discussed in Appendix A. TENG can be divided from bottom to top into copper foil, PDMS (polydimethylsiloxane), the skeleton spacer, and PTFE (polytetrafluoroethylene). The skeleton spacer here is a tic-tac-shaped flexible PDMS skeleton spacer prepared using 3D printing technology. In addition to separating the two triboelectric layers, it also plays the role of isolating signal crosstalk between the TENG units. The fabricated tic-tac-shaped flexible PDMS skeleton spacer is shown in Appendix A. The thickness of the PDMS film is 200 μm, and it is transparent and stretchable. The process of fabricating PDMS with grooves to form microstructures using 3D printing is detailed in Appendix A. In order to determine whether the PDMS film has a microstructure, the surface of the PDMS was observed using a scanning electron microscope, as shown in Appendix A. Meanwhile, we also placed the prepared PDMS film under natural light to observe its light transmission effect. For details, see Appendix A.

To comprehend the foundational principle of current formation in TENGs, one can refer to the atomic scale illustration of charge transference that occurs during the frictional interaction between two materials [40], as depicted in Figure 1c. In a state devoid of friction between two materials situated at a certain distance, the electron clouds surrounding the material surface atoms remain distinct, with no overlapping, as illustrated in Figure 1c(i). During this stage, the electrons of the strong well are bound within a specific orbit by the potential well, precluding any chances of their departure. When an external force acts on the two materials, this leads to a reduced distance between the materials. At this time, the electronegativity of PDMS is weak, and it contributes electrons as a strong well. But the electronegativity of PTFE is strong, so it captures electrons as a weak well. This causes the previously stable single potential well to morph into an asymmetric double potential well. Consequently, the electron clouds around the atoms on the respective material surfaces begin merging. This merging expansion leads to a decrease in the energy barrier between the materials, propelling the electrons from a strong well into a weaker one, and manifesting the triboelectric effect, as shown in Figure 1c(ii). When the two atoms can no longer approach each other at the atomic scale, the electron cloud does not change at this time; therefore, the barriers of the two restore their initial state, as shown in Figure 1c(iii). Upon the removal of the external force, the distance between the materials starts augmenting and the electron cloud overlap area begins reducing. To maintain stability in a single potential well, the previously transitioned electrons revert to their initial positions, as indicated in Figure 1c(iv). This entire process continually recurs, triggering continuous electron motion, and culminating in the generation of a current.

The schematic diagram of the adhesive layer after contact with human tissue is shown in Figure 1d. The LAP nanosheets in the adhesive layer provide a platform for enhanced substrate interactions via electrostatic interaction and hydrogen bonding, which is beneficial for specific flexible bioelectronics that require adhesion and structural integrity. The actual object diagram is shown in Figure 1e. The length, width, and total thickness of the SSFHMI are 95 mm, 55 mm, and 0.75 mm, respectively.

### 3.2. The Hydrogel-Based Adhesive Layer of SSFHMI

The tissue adhesion is critically important for HMI. The softness of hydrogels is favorable in their adhesion to different substrates, which is largely thanks to their incorporated functional groups. The chemical or physical linkages enable the formation between hydrogels and the surrounding substrates [41]. LAP nanosheet-based nanocomposite hydrogels are transparent, stretchable, self-healable, and adhesive [42]. POEGMA-based copolymers synthesized from OEGMA and DEGMA have been reported to develop hydrogels due to their excellent biocompatibility, anti-fouling performance, and ease of preparation [43].

As shown in Figure 2a,b, in this work, LAP nanosheets were used as physically crosslinked epicenters, where a single LAP nanosheet connects to multiple polymeric chains and forms a physically crosslinked network, subsequently forming nanocomposite hydrogel adhesives. In the application of HMI interfaces, the self-healing capability and tensile performance of hydrogels are critically important. The self-healing ability ensures that the HMI interfaces can maintain their integrity and continuous functionality after long-term use or accidental damage. Meanwhile, a good tensile performance can guarantee the HMI interfaces’ stable performance under various pressures. As shown in Figure 2c(i), which shows the tensile stress-strain curves of nanocomposite hydrogel adhesives with an OEGMA/DEGMA ratio (1:1.5) under varied total monomer concentrations. It is worth noting that the OEGMA/DEGMA −1–1.5–25% displays the highest slope at the same tensile degree, which means that its Young’s modulus is the largest, indicating that the adhesive layer made at this ratio has better elasticity and is more suitable for a soft and easy-to-strain skin surface.

Owing to the reversibility of hydrogen bonding and electrostatic interaction, the developed nanocomposite hydrogel adhesives demonstrated an excellent self-healing performance, which was quantitatively determined with dynamic strain amplitude tests. As shown in Figure 2c(ii), the G′ of the nanocomposite hydrogel adhesives is smaller than the G′′ under 800% strain, indicating that the gel network was broken; however, G′ and G″ recovered to their original values within seconds in each cycle when the strain was released to 1%. These results could be explained by the appropriate balance between the mobility of polymer chains and the dynamic crosslinking of the hydrogel networks.

Figure 2d indicates the strong adhesion of the nanocomposite hydrogel adhesives to porcine skin tissues and the FPCB substrate, respectively. Additionally, the adhesive strength of the nanocomposite hydrogel to porcine skin tissues and the FPCB substrate were further characterized by a lap shear test. The results indicate that the synergistic effects of several physical interactions made the nanocomposite hydrogel adhesives display an excellent adhesive performance. The FPCB substrate exhibits a higher adhesion strength than that of porcine skin tissues, which could be due to stronger physical interface interactions, resulting in an increase in adhesion.

### 3.3. Output Performance of a Single TENG

The underlying reason why a TENG generates AC signals can be explained by its working principle at the macro scale. As depicted in Figure 3a, this principle is the result of the combined effect of triboelectricity and electrostatic induction. Upon the application of an external force, the distance between the two triboelectric layers decreases, generating surface charges of opposite signs. The electronegativity of PDMS is weaker than that of PTFE, thus the PDMS layer is positively charged and the PTFE layer is negatively charged [44]. This leads to the formation of opposite charges between the back electrode and the connected PTFE triboelectric layer, causing a flow of electrons from the ground to the back electrode, and thus a current from the back electrode to the ground, as shown in Figure 3a(i). Once the two triboelectric layers are in full contact, the positive and negative charges on the back electrode cancel each other out, eliminating any potential difference and thus stopping the electron flow and current, as shown in Figure 3a(ii). As the external force is removed and the distance between the two triboelectric layers increases, an induced potential difference forms between the back electrode and the ground. This causes the electrons to move from the back electrode to the ground, generating a current from the ground to the back electrode, as shown in Figure 3a(iii). Upon the complete separation of the two triboelectric layers, more charges accumulate on the back electrode, as shown in Figure 3a(iv). This process repeats continuously, with electrons constantly moving and generating equal amounts of current in opposite directions.

The output performance of the TENG is a crucial indicator to determine whether it is suitable as a sensor. Here, we utilized an electrometer and an oscilloscope to conduct open-circuit voltage (Voc), short-circuit current (Isc), and short-circuit charge (Qsc) tests and data recording for TENG, as shown in Figure 3b–d. The tested results reveal an open-circuit voltage of approximately 13 V, a short-circuit current of about 12 nA, and a transferred charge of roughly 9 nC on the TENG. The voltage was used as an input quantity to serve as the trigger source for commanding the terminal. Additionally, we tested the outputs of the TENG when connected with different external loads. As the load resistance increases, the voltage presents an enhancement while the current delivers an opposite variation, as illustrated in Figure 3e. When the external resistance is 50 MΩ, the peak power density value reaches about 0.29 mW/m^2^. This means that the TENG has a relatively large internal resistance and a small output charge density. For HMI applications, it is more suitable to be used as a small-signal self-powered sensor rather than an energy harvester. 

Furthermore, durability serves as an essential criterion for a majority of sensors that warrant comprehensive evaluation. We utilized a linear motor to conduct a fatigue test for the TENG, setting the moving distance at 30 cm, the speed at 500 cm/s, and the acceleration at 2 cm/s^2^. Following 18,000 cycles of contact separation at ƒ = 3 Hz, no significant signal decay was noted, as indicated in Figure 3f. The difference between the positive and negative peaks of the waveform is attributed to the structure of the skeleton spacer and the impact of the linear motor. The nearly equivalent integral area presents that the number of electrons transferred during each cycle is equal, which brings a convincing performance overall in the contact-separation working mode.

### 3.4. Output Characterization of the TENG Array

We also conduct the output performance of the SSFHMI, which means that all the nine TENG units on the device have been tested. A linear motor was used to simulate finger presses on the TENG in the SSFHMI with a fixed working frequency to observe their consistency. As shown in Figure 4a,b, under the same working conditions, the output of the TENGs of the nine channels are nearly identical, providing reassurance for command execution in practical applications. In addition to the above signals, we also conducted a single waveform test for the TENG array at the operating frequency of 5 Hz, as shown in Appendix A.

Moreover, to clarify whether there are differences in the outputs between each two channels due to unknown factors, we performed a two-sample *t*-test and drew a heat map, as illustrated in Figure 4c. The values tested for every two channels on the heat map are close to 1, indicating that there are no obvious differences between two different channels, that is, their values are nearly identical, which more convincingly demonstrates the consistency of the amplitudes of the nine channels.

Furthermore, we conducted interference tests on a single TENG in the device, as presented in Figure 4d. When a single TENG is pressed, although the surrounding TENGs are disturbed and their heat maps turn light blue, their outputs are relatively small and do not reach the trigger threshold we set for executing the commands, so the device is less likely to have misoperation during utilization.

### 3.5. Signal Coding of the Proposed SSFHMI for Intelligent Applications

To explore the potential application scenarios of the proposed SSFHMI in-depth, it is necessary to design the correct action coding and signal processing circuits. The signal coding and the operational process of this device for intelligent applications are demonstrated in Figure 5a, consisting of four parts: SSFHMI as an action encoder, signal processing circuit, microcontroller, and intelligent application terminal. During the operation of the application, it is controlled by the hybrid code groups, which are generated by touching and pressing the TENG array on the SSFHMI. Upon receiving a touch or press from the external environment, the SSFHMI converts it into an electrical signal, and the signal is inputted into the microcontroller unit (MCU). After the signal processing within the MCU, the pre-coding of the TENG behavior, and the comparison with a predefined threshold, it is determined whether it is a signal that triggers an action, and the corresponding effect is displayed on the PC terminal.

First of all, the SSFHMI has a password lock function and requires decoding and verification before use. As displayed in Figure 5b, with numbers 1 to 9 mapped on the device’s TENG array through coding, the platform would generate the same key every 30 s, and only when the users input the correct passwords through the device will they see a successful unlocking prompt, as shown in Figure 5b(i); otherwise, they would see an unlocking failure prompt, preventing access to the next interface as shown in Figure 5b(ii). The device’s anti-misoperation function is an important factor in evaluating its response sensitivity and reliability. To achieve the anti-misoperation function, we set a threshold value of 12 V for the trigger signal. When the output of the TENGs does not reach the threshold, it will not be recognized as an effective signal and cannot trigger the function of inputting numbers. Only when the output of the TENGs reaches the threshold we set will it be recognized as an effective signal, serving as a trigger source to execute the function of inputting numbers, as shown in Figure 5b(iii). Appendix A provides a detailed application demonstration. In addition, the relationship between the pressure and output performance is studied, as shown in Appendix A. When the pressure is relatively low (5 N), the peak output voltage is smaller, about 3.8 V. When the pressure value reaches 15 N, the peak output voltage reaches the maximum value, about 13.7 V. After that, with the increase in pressure, the peak output voltage shows a downward trend, at 30 N, the peak output voltage is about 13.3 V. In order to ensure the rigor of the experiment, we also added a group that at a pressure of 50 N, the output voltage of the TENG also shows a downward trend. The reason for this phenomenon may be that as the pressure increases, the contact separation process of the TENG on the sensing unit is incomplete, resulting in a slight downward trend in the peak output voltage, but the output voltage remains around 13 V.

Next, a medical call system based on this device was developed to perform medical warnings using SSFHMI action codes. It has been designed in several states for the system to work, as displayed in Figure 5c(i), and each TENG on the device is content-coded, such as Hungry, Emergency, Help, Drink, etc. When the corresponding TENG is pressed, the corresponding content is displayed on the terminal, then the healthcare personnel can promptly resolve the issues upon noticing them, as illustrated in Figure 5c(i) and Appendix A. 

Besides, we compared the structure complexity and interference resistance of the HMI interfaces proposed by us with those proposed by other researchers [45,46,47,48,49,50,51,52], as shown in Figure 5c(ii). It is not difficult to see from the figure that the simpler the structure compared with SSFHMI, the poorer its resistance to interference; and the stronger the resistance to interference compared with SSFHMI, the more complex its structure. Therefore, the SSFHMI structure we proposed is relatively simple, has a lower production cost, and has a stronger resistance to interference, making it suitable for HMI interface systems. 

Beyond that, we have also utilized the device to control media players and play the electronic piano keyboard, as demonstrated in Appendix A. The successful realization of the above applications not only proves the excellent multi-functional control ability of the proposed SSFHMI but also broadens the application horizons for its role in intelligent control. Moreover, it is expected to be further miniaturized and arrayed for development. More code combinations will bring more functions, which bring creative ideas to portable and smart electronic devices.

## 4. Conclusions

In this work, we report a self-powered, skin adhesive, and flexible human–machine interface (SSFHMI) for wearable and multifunctional HMI application. Compared with the previously reported TENG-based HMI interface, the simplistic fabrication process, efficacious adherence, and a skeleton spacer of the proposed SSFHMI not only results in a stable signal but also consistently delivers a reliable performance. A single TENG unit fabricated on the SSFHMI is capable of reaching an output voltage of 13 V and a short-circuit current of 12 nA, and maintaining an excellent durability after 18,000 cycles of work. Moreover, a hydrogel is synthesized to serve as an adhesive interface, the adhesive strength reaches up to 20 kPa between the SSFHMI and the tissue, ensuring a strong adhesion between the device and the complex human body surface. In order to prevent the electrical signals interfering with each other, we prepared a flexible skeleton spacer. Finally, by integrating with an MCU, the SSFHMI can also be used as a programmable touch operation platform that supports several typical application scenarios, including medical calling, music media playback, security unlocking, and electronic piano playing.

This work feature is not only the implementation of manufacturing techniques allied with commonplace materials fosters ease in miniaturization and array evolution, but also its methodology’s feasibility and application potential. But materials and structures need to be more sophisticatedly designed to achieve a higher output performance and durability. Future work will involve introducing other novel materials and structure designs to achieve higher stability, durability, and ease of customization. This SSFHMI is suitable for tremendous potential application scenarios. It could prove instrumental in various domains, including 3D control, Virtual Reality/Augmented Reality (VR/AR), advanced security systems, and portable electronic devices, among others.

## Figures and Tables

**Figure 1 nanomaterials-14-01365-f001:**
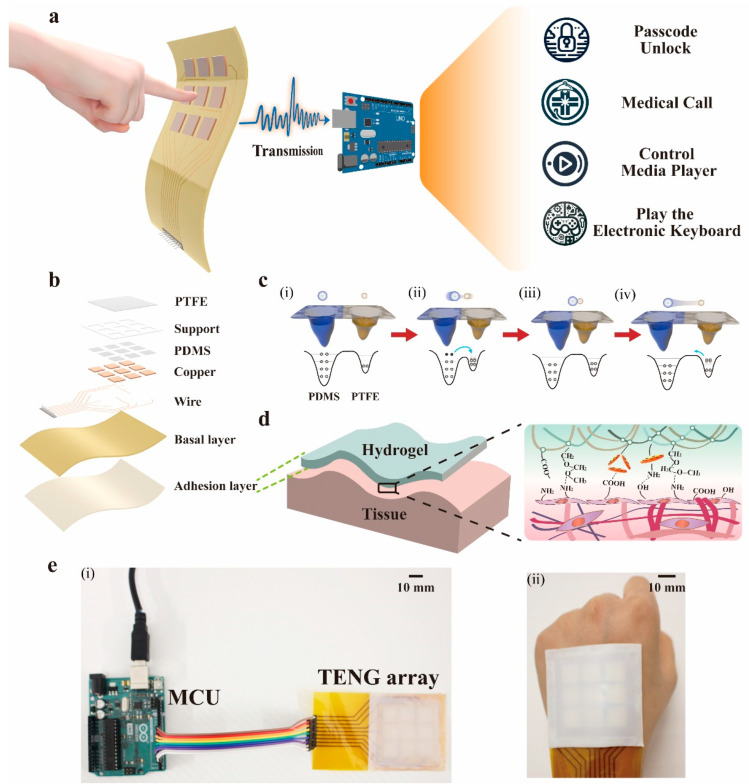
Overview diagram of the proposed SSFHMI. (**a**) The proposed SSFHMI and its application in intelligence interaction, medical calls, media player control, password unlocking, and electronic keyboard playing. (**b**) Construction of the SSFHMI. (**c**) Atomic-scale and macroscopic charge transfer mechanisms during friction between PDMS and PTFE. (**i**) PDMS and PTFE are in Separated stage. (**ii**) Compressing stage. (**iii**) Compressed stage. (**iv**) Separating stage. (**d**) The diagram of the mechanism of adhesion effect after contact between the adhesive layer and the human tissue. (**e**) Photograph of the proposed SSFHMI. (**i**) MCU connected with SSFHMI. (**ii**) SSFHMI attached to the back of the hand.

**Figure 2 nanomaterials-14-01365-f002:**
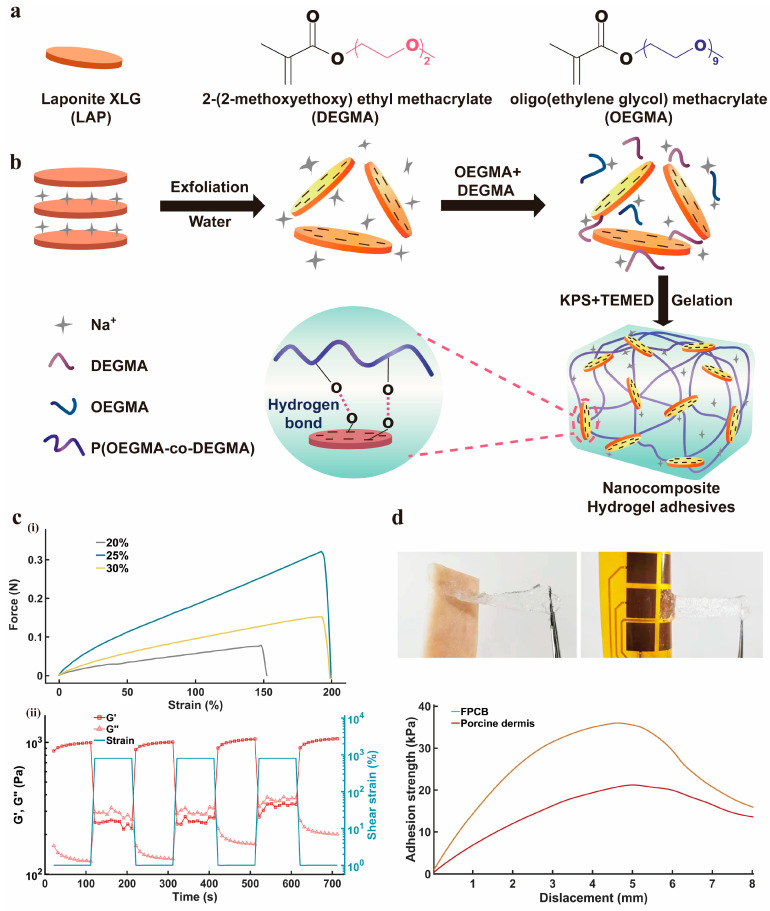
Preparation, properties of the hydrogel-based adhesive layer. (**a**) Schematic illustration of the material used to prepare the adhesive layer. (**b**) Steps and methods for preparing the adhesive layer. (**c**) The tensile stress-strain curves and dynamic strain amplitude curves of the nanocomposite hydrogel adhesives. (**i**) Tensile stress-strain curves of LAP nanosheets at different concentrations. (**ii**) Self-healing characteristic curve. (**d**) Photograph of the adhesive layer for bonding SSFHMI to biological tissues, and the viscosity curve of the adhesive layer.

**Figure 3 nanomaterials-14-01365-f003:**
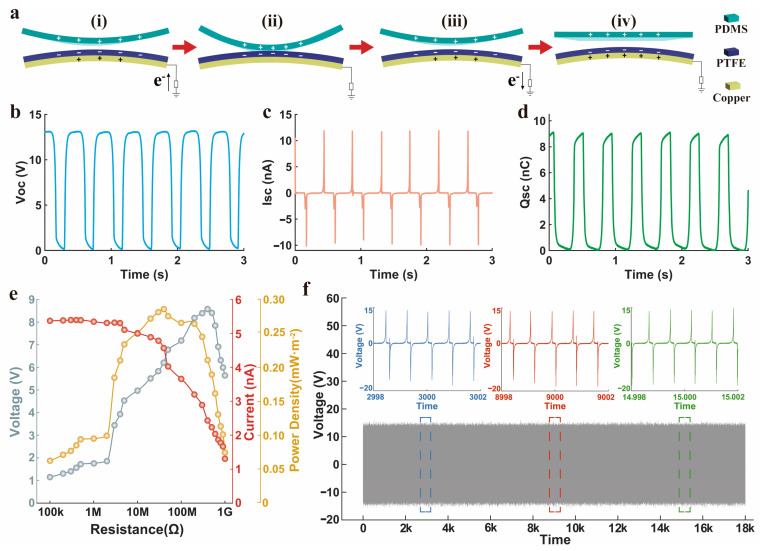
Electrical characterization of the TENG. (**a**) Illustration of the macro-level working mechanism of a single-electrode TENG. (**i**) Compressing. (**ii**) Compressed. (**iii**) Separating. (**iv**) Separated. (**b**–**d**) Open-circuit voltage, short-circuit current, and short-circuit charge at the working frequency of 3 Hz, respectively. (**e**) Output voltage, current, and power density under the different external load resistances. (**f**) Output voltage of the TENG during 18,000 working cycles.

**Figure 4 nanomaterials-14-01365-f004:**
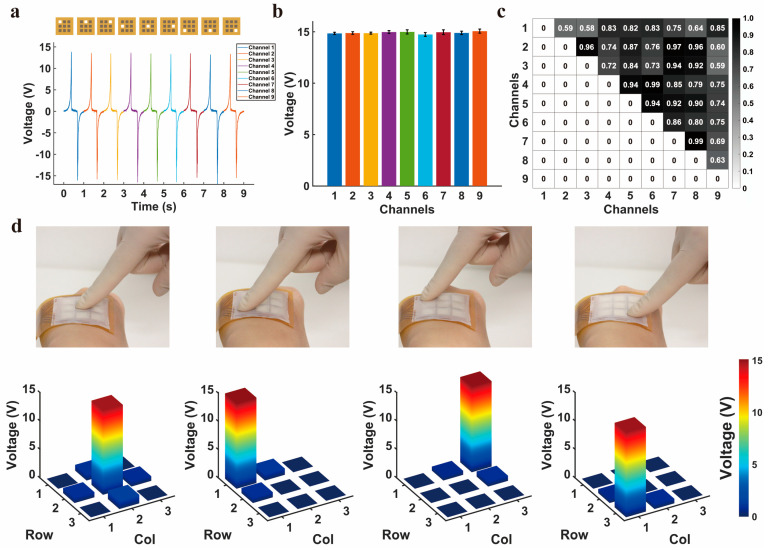
The output characterization of the TENG array on the proposed SSFHMI. (**a**) The output waveform of nine TENGs. (**b**) The output voltage of nine TENGs. (**c**) Results of the two-sample T-test performed between TENGs. (**d**) 3D heat map corresponding to the different pressing positions on the TENG array.

**Figure 5 nanomaterials-14-01365-f005:**
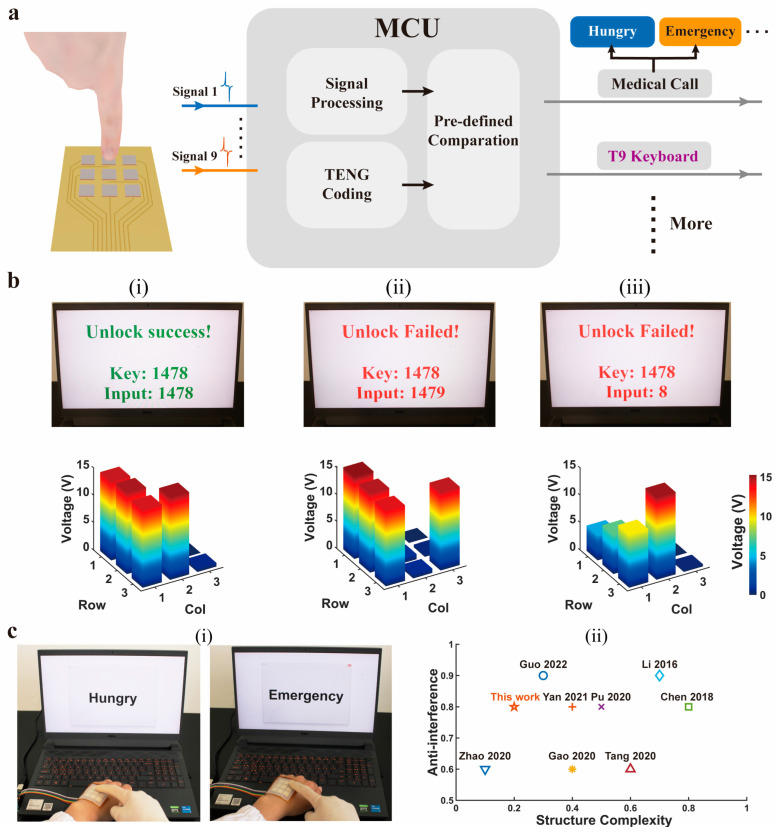
The signal coding of the proposed SSFHMI for intelligent control. (**a**) Schematic diagram of the human–computer interaction using SSFHMI. (**b**) Password lock function of the T9 keyboard and its 3D heat map of corresponding operations. (**c**) Application of the medical call. (**i**) Practical application demonstration. (**ii**) Comparison with similar work in Structure Complexity and Anti-interference [45,46,47,48,49,50,51,52].

## Data Availability

Data are contained within the article or Appendix A.

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
