# Peer review of "A Self-Powered, Skin Adhesive, and Flexible Human–Machine Interface Based on Triboelectric Nanogenerator"

_nanomaterials, 2024, doi:10.3390/nano14161365_

Round 1

Reviewer 1 Report

Comments and Suggestions for Authors

In this manuscript, the authors report a self-powered, skin-adhesive, and flexible human-machine interface (SSFHMI) based on a triboelectric nanogenerator for wearable human-machine interaction. They also synthesized a hydrogel to serve as an adhesive interface for the SSFHMI. The authors demonstrated the application of SSFHMI by integrating it with a microcontroller for various uses. Therefore, I recommend this article for publication in the Journal of Nanomaterials. However, before publishing, the authors need to discuss or add additional data addressing the following questions. The following comments are provided with the hope of further improving the overall quality of the work.

1.  The authors mentioned that they prepared a PDMS film with microstructures. However, no evidence is provided for this. The authors should provide evidence of the microstructures in the PDMS film.

2.    The authors used PDMS and PTFE to fabricate the TENG. It is well known that both are triboelectric negative materials. Why did the authors use two negative triboelectric materials for their study? Additionally, the authors provided a general mechanism for TENG operation without specifying which material donates electrons and which one accepts electrons during operation, leading to confusion for readers. The authors should provide a clear and detailed mechanism for the TENG operation based on the materials used.

3.   In addition, fluoropolymer PTFE has a more triboelectric negative nature than PDMS, indicating that PTFE is more likely to attain a negative charge compared to PDMS (10.1002/adfm.202004714). However, the authors mentioned this incorrectly in the mechanism (Figure 3a). The authors should modify the working mechanism and explain accordingly.

4.  Why the triboelectric output signal shape is different as seen in Figures 3b and 3f. Authors should maintain constancy in data.

Comments on the Quality of English Language

Typo errors should be corrected

Reviewer 2 Report

Comments and Suggestions for Authors

This manuscript reports a self-powered, skin adhesive, and flexible human-machine interface (SSFHMI) for wearable and multifunctional HMI application, with detailed investigation in material, device and system integration, which could be of interest to a broad readership in this field. But before considering for publication, there are some comments that the authors should address first.

1.       What is the total thickness of the HMI consisting multiple layers? It looks a bit thick and how to ensure the wearing comfort of it on skin.

2.       Why choose PDMS and PTFE as the opposite triboelectric layers, since they are both considered as negative triboelectric materials in most applications? When using a more positive material such as copper itself as the positive triboelectric material, the output could be further improved.

3.       Please add a scale bar for the photos in Fig. 1.

4.       Since the adhesive layer is directly contact with human skin, so how is its biocompatibility and skin irritation effect?

5.       In Fig. 3e, as the resistance continuously increases, why does the voltage show a decreasing trend after 100 MΩ?

6.       The voltage waveform in Fig. 3f is quite different from that in Fig. 3b, how is it measured?

7.       How is the output dependency on applied pressure or force, this should be also investigated.

Comments on the Quality of English Language

Minor revision of English is required.

Reviewer 3 Report

Comments and Suggestions for Authors

In the manuscript, a self-powered, skin adhesive, and flexible human-machine interface based on a triboelectric nanogenerator has been reported. This work is interesting, and minor changes are required before publication in this journal. 

1) The introduction should be a more comprehensive and detailed discussion of the fundamentals of state-of-the-art

2) Authors should consider adding new recently reported relevant articles to strengthen the proposed work: 10.1016/j.talanta.2024.125817; 10.20517/ss.2023.54

3) Authors should provide a summary comparison table on recently reported literature with several parameters including LOD, sensitivity, dynamic range, reproducibility, and application

4) Please mention the challenges and limitations

5) How the sensing region was fabricated is unclear and What about crosstalk on the data lines?

Comments on the Quality of English Language

In the manuscript, a self-powered, skin adhesive, and flexible human-machine interface based on a triboelectric nanogenerator has been reported. This work is interesting, and minor changes are required before publication in this journal. 

1) The introduction should be a more comprehensive and detailed discussion of the fundamentals of state-of-the-art

2) Authors should consider adding new recently reported relevant articles to strengthen the proposed work: 10.1016/j.talanta.2024.125817; 10.20517/ss.2023.54

3) Authors should provide a summary comparison table on recently reported literature with several parameters including LOD, sensitivity, dynamic range, reproducibility, and application

4) Please mention the challenges and limitations

5) How the sensing region was fabricated is unclear and What about crosstalk on the data lines?

Round 2

Reviewer 1 Report

Comments and Suggestions for Authors

The authors have responded to all the questions appropriately. 

Author Response

Comments and Suggestions for Authors: The authors have responded to all the questions appropriately. Response: Thank you very much for reviewing our manuscript and put forward valuable and specific revision suggestions, we sincerely appreciate your serious and rigorous work. We are very pleased that our revised manuscript meets the level of your requirements.

Reviewer 2 Report

Comments and Suggestions for Authors

The authors have addressed most of the comments, but the output dependency on applied pressure should be added under practical usage scenarios.
